# Human Papillary and Reticular Fibroblasts Show Distinct Functions on Tumor Behavior in 3D-Organotypic Cultures Mimicking Melanoma and HNSCC

**DOI:** 10.3390/ijms231911651

**Published:** 2022-10-01

**Authors:** Shidi Wu, Marion Rietveld, Marieke Hogervorst, Frank de Gruijl, Sjoerd van der Burg, Maarten Vermeer, Remco van Doorn, Marij Welters, Abdoelwaheb El Ghalbzouri

**Affiliations:** 1Department of Dermatology, Leiden University Medical Center, 2333 ZA Leiden, The Netherlands; 2Department of Medical Oncology, Oncode Institude, Leiden University Medical Center, 2333 ZA Leiden, The Netherlands

**Keywords:** papillary fibroblast, reticular fibroblast, cancer-associated fibroblast, melanoma, HNSCC, EMT, tumor invasion, tumor microenvironment

## Abstract

Human dermis can be morphologically divided into the upper papillary and lower reticular dermis. Previously, we demonstrated that papillary (PFs) and reticular (RFs) fibroblasts show distinct morphology and gene expression profiles. Moreover, they differently affect tumor invasion and epithelial-to-mesenchymal transition (EMT) in in vitro 3D-organotypic cultures of cutaneous squamous cell carcinoma (cSCC). In this study, we examined if these distinct effects of PFs and RFs can be extrapolated in other epithelial/non-epithelial tumors such as melanoma and head and neck squamous cell carcinoma (HNSCC). To this end, 3D-Full-Thickness Models (FTMs) were established from melanoma (AN and M14) or HNSCC cell lines (UM-SCC19 and UM-SCC47) together with either PFs or RFs in the dermis. The interplay between tumor cells and different fibroblasts was investigated. We observed that all the tested tumor cell lines showed significantly stronger invasion in RF-FTMs compared to PF-FTMs. In addition, RF-FTMs demonstrated more tumor cell proliferation, EMT induction and basement membrane disruption. Interestingly, RFs started to express the cancer-associated fibroblast (CAF) biomarker α-SMA, indicating reciprocal interactions eventuating in the transition of RFs to CAFs. Collectively, in the melanoma and HNSCC FTMs, interaction of RFs with tumor cells promoted EMT and invasion, which was accompanied by differentiation of RFs to CAFs.

## 1. Introduction

Cancer development and progression does not solely rely on the tumor cell intrinsic alterations [1]. For the last decades, the tumor microenvironment (TME) has been recognized as a vital constituent for promoting events including tumor growth, invasion and metastasis, as well as mediating therapeutic resistance in carcinogenesis ever since the “seed and soil” theory was proposed by Paget in 1889 [2]. Therefore, microenvironment-targeted strategies have been developed to improve cancer therapy response. To date, although remaining at a relatively early stage, targeting TME compartments showed satisfying efficiency in preclinical phase [3]. The TME comprises a variety of heterogeneous cell types including fibroblasts, endothelial cells, and immune cells. This kind of supporting niche is crucial for nurturing the surrounding tumor cells which results in malignant behavior and cancer progression [4]. One mechanism by which TME facilitates tumor invasion and metastasis is inducing the tumor cells to undergo epithelial-to-mesenchymal transition (EMT).

EMT is a well-defined program that reversibly turns epithelial cells into quasi-mesenchymal cell states [5]. During this process, the activation of EMT causes the loss of cell polarity, disruption of tight cell–cell junctions and degradation of the underlying basal membrane, which allows tumor cells to invade into the adjacent stroma and disseminate to distant tissues [6]. The hallmark of an EMT occurrence is the reduction in E-cadherin, repressed by the EMT-inducing transcriptional factors including mainly the TWIST, SNAIL and ZEB families [7,8], and the increased expression of mesenchymal proteins such as vimentin and N-cadherin. The induction of EMT by TME has been discovered long before. In tumorigenesis, cancer cells can generate a microenvironment, which further favors the growth and invasion for themselves. However, the TME also adjusts its morphological characteristics, which yields an increase in inflammatory cell infiltration and the development of cancer-associated fibroblasts (CAFs) [9], which further induces the EMT program in carcinoma cells.

Human dermis comprises two layers, namely the papillary and reticular dermis, which show diversities in location, extracellular matrix (ECM) composition and organization, fibroblast morphology and function [10]. As the dermis lies immediately beneath the epidermis (300–400 μm), the papillary dermis is thin and characterized by poorly organized matrix fibers and a relatively high cell density while the underlying reticular dermis is demarcated from the papillary dermis by a vascular plexus, which is much thicker and characterized by well-organized matrix fibers and a low cell density [11]. In terms of the ECM components, the papillary dermis contains a higher ratio of collagen type III to I, expresses more decorin, collagen type XII, collagen type XVI, tenascin-C and less versican compared to the reticular dermis [12,13,14]. Moreover, the fibroblast heterogeneity in these two dermal layers was also investigated in our previous studies, indicating that PFs and RFs have distinct gene expression profiles and affect epidermal regeneration differently [15,16]. Furthermore, PFs can gain an RF phenotype after prolonged culture or TGFβ induction [17]. The ECM produced by these two fibroblast subtypes also differed and showed dissimilar effects on skin homeostasis [18]. In addition, we also firstly identified their roles in cSCC invasion and EMT [19] by using in vitro 3D-FTMs. However, whether these two fibroblast subtypes universally play a distinct role in tumor progression in other cancer types is not known.

In this study, we generated melanoma (cell lines AN and M14) and HNSCC (cell lines UM-SCC19 and UM-SCC47) FTMs that harbor PFs (PF-FTMs) or RFs (RF-FTMs) and further examined the regulatory effects of these two fibroblast subtypes in tumor invasion, followed by the investigation on tumor epidermal cell proliferation, activation, differentiation and EMT. Furthermore, the effect of tumor cells on the differentiation of fibroblast subpopulations was studied.

## 2. Results

### 2.1. Identifying Potential Biomarkers for PFs and RFs

Expanding on our previous studies of PF and RF transcriptomes, we established multiple mRNA and protein biomarkers specific for the respective dermal fibroblast subtype. As shown in Figure 1a, the papillary dermis is composed of loose connective tissue, whereas the reticular dermis is densely packed with matrix fibers. The two fibroblast subtypes were isolated and cultured from these two layers. Subsequently their expression profile on both mRNA and protein level in a 2D monolayer culture was measured and compared (Figure 1b–f). PFs showed a higher expression of the genes including CLEC3B, COL7A1, MMP12 and MMP3 (* *p* < 0.05) when compared to RFs. The gene expression of CLEC3B was further validated on the protein level (Figure 1d,e) in six donors, and a significantly higher expression of this protein was observed in PFs compared to RFs. RFs demonstrated a higher expression of genes including COL11A1 (** *p* < 0.01), HAS1 (* *p* < 0.05), ITGβ2 (** *p* < 0.01), MCAM, VCAM, POSTN and SPP1. Interestingly, COL11A1 was absent or very weakly expressed in PFs. This observation was validated in 3D-FTMs composed of healthy primary keratinocytes and fibroblasts. Cells positive for COL11A1 were only detected below the epidermis in RF-FTMs as indicated by arrows (Figure 1f).

### 2.2. PFs and RFs Have a Dissimilar Effect on Invasive Behavior of Melanoma and HNSCC Cells in 3D-FTMs

Next, we examined the effects of PFs and RFs on the invasive behavior of 3D tumor FTMs mimicking melanoma and HNSCC. To this end, we generated PF-containing and RF-containing FTMs and performed HE staining (Figure 2a) followed by calculating the percentage of the tumor ingrowth area in the dermis (Figure 2b). PF-FTMs, onto which different tumor cell lines AN, M14, UM-SCC19 or UM-SCC47 were seeded, showed no to slight invasion into the dermal compartment, while the area and number of invaded cell islands was apparently higher in reticular dermis, especially in the melanoma FTMs (M14, * *p* < 0.05; AN, ** *p* < 0.01).

### 2.3. PFs and RFs Differently Contribute to Tumor Morphogenesis in Melanoma and HNSCC FTMs

Using immunohistochemistry, we subsequently characterized the expression profiles of biomarkers for tumor cell activation, proliferation and differentiation, and basement membrane (BM) formation in all the generated FTMs.

We firstly observed the expression pattern of two melanoma markers NKI/beteb and SOX10 in all melanoma FTMs. As shown in Figure 3a,b and Appendix A, the PF-FTMs demonstrated very little NKI/beteb positive cells when compared to the RF-FTMs. Similarly, the RFs up-regulated the expression of SOX10 as the mRNA level of SOX10 was approximately 2.5 and 11 times higher in the AN-RF-FTMs (* *p* < 0.05) and M14-RF-FTMs (*** *p* < 0.001), respectively, compared to the corresponding PF-FTMs (Appendix A). Next, we examined to what extent PFs and RFs affected HNSCC cell activation by detecting the expression of keratins (K) 17. As demonstrated in Figure 3c,d and Appendix A, K17 was expressed in all HNSCC FTMs regardless of the fibroblast type in the dermal compartment. However, the expression was more confined in the UM-SCC47-RF-FTMs than UM-SCC47-PF-FTMs. An intact BM is crucial for preventing tumor invasion, since it functions as a mechanical barrier for hindering the tumor cells invading into the dermis and further tissues. The BM formation and its integrity were assessed by detecting the expression of Collagen Type IV (COL4), which is well recognized as one of the major components of BM. In the PF-FTMs, COL4 displayed a proper expression pattern between the epidermis and dermis, while in the RF-FTMs a disruption of the BM barrier was noticed, resulting in the BM lining the invasion area in the dermal compartment. By comparing the numbers of Ki67-positive cells between the PF-FTMs and RF-FTMs, we further explored whether the fibroblast heterogeneity differentially regulated the tumor cell proliferation ability. An increase in Ki67-positive cells was observed in both the RF-FTMs with AN (*** *p* < 0.001; Figure 3a,e) and M14 (*** *p* < 0.001; Figure 3b,f) epidermis compared to that in the PF-FTMs, while such a difference was not profound in the HNSCC FTMs. The expression of the early epidermal differentiation marker K10 showed no differences among the different FTMs, indicating a proper execution of this early differentiation program (Figure 3a–d and Appendix A).

### 2.4. RFs Induce the Expression of EMT-Related Genes in Melanoma and HNSCC FTMs

To unravel the mechanism of the distinct regulatory effects of PFs and RFs on tumor invasion, the gene expression analysis of a list of 84 EMT-related markers was performed in both melanoma and HNSCC FTMs by using the EMT PCR array kit from Qiagen. These genes were demonstrated to either change their own expression or regulate those genes whose expression is altered during the EMT process, and part of which are presented in Figure 4. For the melanoma FTMs, as shown in Figure 4a, most of the EMT-inducing genes were significantly more expressed in the RF-FTMs compared to that in the PF-FTMs including TGFβ3 (* *p* < 0.05 in AN), SPARC (* *p* < 0.05 in both AN and M14), ITGα5 (* *p* < 0.05 in M14), SPP1 (* *p* < 0.05 in M14), TMEFF1 (* *p* < 0.05 in M14), ZEB2 (* *p* < 0.05 in M14) and FZD7 (*** *p* < 0.001 in M14). In the HNSCC FTMs shown in Figure 4b, a similar pattern was found as the majority of the EMT-inducing genes were up-regulated in the RF-FTMs as well when compared with the PF-FTMs, with N-cadherin (* *p* < 0.05 in SCC19), SPARC (** *p* < 0.01 in SCC47), TGFβ3 (* *p* < 0.05 in SCC47; ** *p* < 0.01 in SCC19), TMEFF1 (* *p* < 0.05 in SCC47) and FZD7 (* *p* < 0.05 in SCC19) being statistically significant. Moreover, SerpinE1, a member from the Serpin Family, which is overexpressed in HNSCC and tightly correlated with tumor budding and metastatic growth in lymph nodes, was two times higher in the RF-FTMs than in the PF-FTMs. However, the expression of SerpinB4, a well-known biomarker for SCC, did not show a difference between the PF- and RF-HNSCC FTMs. Interestingly, MMP9, a metalloproteinase responsible for BM breakdown in tumor invasion, showed a slight downregulation in the melanoma RF-FTMs compared to that in the PF-FTMs and little difference was found between the two types of HNSCC FTMs.

Vimentin, a frequently used marker for EMT, showed increased expression in the RF-FTMs, especially in both the AN and M14 melanoma FTMs as a strong positive staining throughout the invasive area in the reticular dermal compartments was observed (Figure 5a,b). In the HNSCC RF-FTMs, positive staining was presented within the epidermis and in the upper dermis where the tumor cells invaded, while little staining was observed in the PF-FTMs. Both the EMT-inducing transcription factors ZEB1 and ZEB2 were upregulated in the UM-SCC47-RF-FTMs as indicated by arrows, compared to that in the UM-SCC47-PF-FTMs, while ZEB2 expression remained similar in the UM-SCC19-PF-FTMs and UM-SCC19-RF-FTMs. In the melanoma FTMs, RFs did not affect ZEB1 expression but clearly induced ZEB2 protein expression at the ingrowth areas compared to the PFs. Additionally, SPARC, an identified EMT marker that promotes EMT in tumorigenesis by regulating TGFβ signaling, was predominantly expressed within the invaded cell islands with stronger expression in both the melanoma and HNSCC RF-FTMs.

### 2.5. RFs Are the Precursor of CAFs

α-SMA has been recognized as a classic marker for myofibroblasts and CAFs. First, in Figure 6a, by co-culturing the PFs or RFs with melanoma and HNSCC cell lines on a monolayer, we noticed a significant up-regulation of α-SMA expression in the RFs after being co-cultured with AN (** *p* < 0.01), while no such effect was observed in the PFs. Both the HNSCC cell lines did not seem to affect α-SMA expression in the PFs and RFs. Moreover, the melanoma cell lines AN and M14 did not alter the phenotype of the PFs as the expression of the two PF markers CLEC3B and CCRL1 showed little change in the PFs but decreased the RF marker (MGP) expression in the RFs (*** *p* < 0.001 for both AN and M14). As for the HNSCC cell lines UM-SCC19 and UM-SCC47, they showed the opposite effect by downregulating the expression of CLEC3B and CCRL1 in the PFs (*** *p* < 0.001) after tumor cell-fibroblast co-culturing while not affecting the RF marker expression of TGM2 and MGP in the RFs. Interestingly, in our 3D organotypic culture compared to the 2D co-culture, the expression of α-SMA was increased in all the tumor RF-FTMs (Figure 6b), indicating that cell–cell direct interaction or other mechanisms might be involved in the RF-to-CAF differentiation process.

## 3. Discussion

It has been long shown that PFs and RFs are morphologically and physiologically different [20,21,22,23]. Under in vitro culture, PFs have a lean, spindle-shaped appearance while RFs have a squarer and stretched morphology [15]. The difference could also be found in the proliferation rate [18,20] and the production of matrix [24,25] and growth factors [26] between these two groups of fibroblasts. To dissect the heterogeneity of PFs and RFs, efforts have been made to explore specific markers for both fibroblast subtypes. In our previous research we identified several biomarkers for PFs (e.g., PDPN, CCRL1) and for RFs (e.g., MGP and TGM2) [15]. As addition, in this study we enriched our set with potential biomarkers such as CLEC3B, COL11A1, HAS1 and ITGβ2. CLEC3B is known to encode tetranectin, a transmembrane Ca^2+^-binding protein playing an important role in extracellular proteolysis, which is associated with tumorigenesis [27,28]. It is reported that CLEC3B is downregulated in lung cancer and involved in immune activation and proliferation inhibition [29]. Similarly, downregulation of CLEC3B was also reported in several other carcinomas including hepatocellular carcinoma [28] and oral squamous cell carcinoma [30]. In fibroblasts, we observed a consistently higher expression of CLEC3B in the PFs than in the RFs. The antagonistic role of CLEC3B to tumor invasion seems to correspond to our finding that tumor ingrowth was hampered when the underlying dermis was constructed with PFs. To what extend this anti-tumor effect of PFs is due to the high CLEC3B expression requires further investigation. COL11A1 was recently identified as a unique CAF marker [31,32]. High COL11A1 expression level is frequently correlated with tumor invasion, recurrence and poor survival in various carcinomas [33,34,35,36,37]. In our findings, this biomarker was predominantly expressed in the RFs and barely detectable in the PFs, indicating a plausible relatively “pre-activated” status of this fibroblast subtype for a transition to CAFs. HAS1 is one of the three hyaluronic acid synthases which usually synthesize hyaluronic acid (HA). The association between a high expression of HAS1 with tumor metastasis and poor patient survival has been documented in bladder cancer [38,39]. The higher level of HAS1 expressed in the RFs compared to the PFs might be one of the reasons that the RFs favor tumor ingrowth. Surprisingly, our gene analysis showed a much higher MMP3 expression in the PFs than in the RFs. Since it was reported that MMP3 directly cleaves the extracellular domain of E-cadherin, encouraging EMT and cancer-cell invasiveness in breast cancer [40], whether there exists a subtle link between the seemingly protective effect of PFs in tumor invasion and the higher expression of MMP3 should be further investigated.

The distinct effect of PFs and RFs on cutaneous SCC ingrowth in our 3D FTMs was described in our previous research [19]. In this study, we found similar observations for melanoma and HNSCC FTMs. RFs clearly promote tumor invasion while PFs do not, as judged by the measurement of tumor ingrowth and expression of specific biomarkers. The antibody NKI/beteb, which recognizes glycoproteins, has been used as a successful diagnostic marker for melanoma [41]. It is normally expressed at low levels in quiescent adult melanocytes but is overexpressed by proliferating neonatal melanocytes and during tumor invasion. SOX10, another well-known melanoma marker is an indispensable protein for melanoma initiation and maintenance, which was used as a marker for metastatic melanoma [42,43]. In our melanoma RF-FTMs, both markers demonstrated a significantly higher expression in RF-FTMs, indicating the melanoma cell proliferation was greatly induced by the RFs compared to the PFs. The proliferation marker Ki67 confirmed that the most active proliferating cells were predominantly present in melanoma-RF-FTMs. However, little difference was noticed when comparing Ki67 expression between the HNSCC-PF-FTMs and HNSCC-RF-FTMs.

The BM is a dense, 100–300 nm thick lamina that underlies all epithelia. It plays an undoubtful crucial role in normal tissue-invasive programs in morphogenesis, immune surveillance and also in neoplastic events [1,44,45,46]. One of the major components of BM is COL4. By detecting the expression pattern of COL4, we observed that in the melanoma PF-FTMs, COL4 was evenly distributed along the BM in a line, properly separating the tumor epidermis and the papillary dermis. In contrast, the BM was greatly disrupted and encircled the invasion area in the melanoma FTMs with the reticular dermis, indicating a better invasive behavior was obtained by the melanoma cells after interacting with the RFs instead of the PFs. In the HNSCC FTMs, this phenomenon was less significant as we saw disruption regardless of the dermis type, which was consistent with the observation where we also noticed less significance in the ingrowth difference between the HNSCC PF-FTMs and HNSCC RF-FTMs. Since HNSCC is derived from mucosal epithelium, this might be due to the lesser effect of fibroblasts on non-skin-related tumor cells. Nevertheless, the underlying mechanism which is responsible for this discrepancy in different tumor types and tumor cell lines should be further investigated.

A list of genes closely related to EMT were selected and their expression was detected at both the mRNA and protein level. For the EMT transcription factors, both ZEB1 and ZEB2 showed overexpression in the RF models, especially ZEB2 expression in the melanoma-RF-FTMs. It was reported that ZEB1 expression is associated with an invasive phenotype while ZEB2 expression is required for the proliferative melanoma cell state [47,48]. It is clear that the RFs induced both proliferation and invasion in our melanoma FTMs. SPARC is identified as a pro-tumorigenic protein in melanoma [49], HNSCC [50] and various other cancers [51,52,53], mediating tumor growth, invasion and metastasis via ECM remodeling and EMT induction. An obvious overexpression of SPARC was also observed in our RF-FTMs compared to the PF-FTMs. Similarly, staining from another well-studied EMT marker vimentin showed significantly increased expression especially in the cell islands invaded in the reticular dermis. TGFβ signaling is known to induce EMT by upregulating SNAIL, ZEB and TWIST expression via the SMAD-dependent way or by activating ERK in an SMAD-independent fashion [54,55]. Consistent with their binding to the same receptor complexes, TGFβ1, TGFβ2 and TGFβ3 share the capacity to induce EMT. In our control melanoma FTMs, TGFβ3 already showed a greater expression in the RF FTMs, indicating an intrinsically more activated TGFβ signaling in the RFs than in the PFs. This might also partially explain why the RFs, instead of the PFs, induce EMT in our melanoma and HNSCC FTMs.

α-SMA was identified as one of the most frequently used CAF markers [56]. First, in a monolayer culture, we observed the phenotype of the PFs and RFs after being co-cultured with melanoma or HNSCC cells. In the melanoma cell–fibroblast co-culture system, on the one hand, the PF phenotype did not seem to be affected, and on the other hand, a loss of the RF marker and a gain of α-SMA expression was demonstrated in the RFs, especially in the AN melanoma cell co-culture group, indicating a potential CAF transition. In the HNSCC cell–fibroblast co-culture system, an opposite observation was noticed as the expression of PF markers were strongly decreased in the PFs while neither the loss of the RF marker nor the gain of the CAF marker were determined. This might also partially explain why the distinct effect of these two fibroblast subtypes on tumor behavior was less profound in our HNSCC FTMs. Strikingly, in our 3D organotypic culture system, little α-SMA expression was observed in our normal PF or RF FTMs, and after interaction with the melanoma or HNSCC cell lines, the RFs started to express a much higher level of α-SMA, indicating signaling being activated between the tumor cells and the fibroblasts in the tumor stroma. Similar observations were also noticed in our previous cSCC FTMs. The fact that in the 3D culture all the tumor cell lines induced α-SMA expression in the RFs compared to the 2D co-culture indicated that direct cell–cell interaction or other mechanisms might play a crucial role in the CAF activation process, which requires further study. Nevertheless, it is rather clear that RFs instead of PFs might be the precursory CAFs, and under certain activation or interactions with tumor cells, RFs acquire a CAF phenotype and promote tumor invasion. However, the use of α-SMA as a specific CAF marker is also hampered due to its low expression in inflammatory CAFs [57]. To verify the potential differentiation process of RFs to CAFs, more CAF markers should be examined and the mechanisms behind the transition also need to be studied in more detail.

## 4. Materials and Methods

### 4.1. Cell Culture

PFs and RFs were isolated from female (age 30 to 45 years) surplus abdominal skin as previously described [13,26]. Briefly, after disinfection, a depth of 100–300 μm and a minimum of 700 μm of the dermis were dissected to obtain the different dermal layers, respectively, by using a dermatome. Fibroblasts from these two layers were isolated by treating with a cocktail of collagenase (Gibco, Bleiswijk, The Netherlands) and Dispase II (Roche Diagnostics, Almere, The Netherlands) in a 3:1 ratio for 2 h at 37 °C. Then, the cells were filtered through a 70 μm cell strainer and cultured in a standard fibroblast medium at 37 °C and 5% CO_2_. The standard fibroblast medium consisted of Dulbecco’s modified Eagle’s medium (DMEM, Gibco), 5% foetal bovine serum (FBS, Greiner, Alphen aan den Rijn, The Netherlands) and 1% penicillin/streptomycin (P/S, Invitrogen, Breda, The Netherlands).

Normal human epidermal keratinocytes (NHEKs) were also isolated from female surplus mamma reduction skin. In brief, after overnight incubation with Dispase II (Roche Diagnostics), the epidermis was separated from the dermis and then further incubated with 0.05% trypsin to acquire the keratinocytes. Cells were cultured in standard keratinocyte medium consisting of DMEM and Ham’s F12 (Gibco) at a 3:1 ratio, supplemented with 5% FBS (Bioclot, Aidenbach, Germany), 1.1 µM hydrocortisone, 1 µM isoproterenol, 0.087 µM insulin (Sigma-Aldrich, Zwijndrecht, the Netherlands) and 1% P/S. All healthy donors used to isolate primary cutaneous cells were obtained in accordance with the Dutch Law on Medical Treatment Agreement and the Code for Proper Use of Human Tissue of the Dutch Federation of Biomedical Scientific Societies. The Declaration of Helsinki principles were followed when working with human tissue.

Human melanoma cell line AN (skin metastasis) was kindly provided by H. Randolph Byers (Harvard Medical School, Boston, MA, USA) and the M14 cell line (lymph node metastasis) was kindly provided by Kenneth L. Scott (Dana Farber Cancer Institute, Boston, MA, USA). The cell lines were cultured in standard fibroblast medium at 37 °C with 5% CO_2_. HNSCC cell lines (UM-SCC19 and UM-SCC47) were obtained from the University of Michigan (Ann Arbor, MI, USA) and cultured in standard keratinocyte medium at 37 °C with 7.3% CO_2_.

### 4.2. 2D Co-Culture

By using 6-well Transwell system (Costar, #3450), 4 × 10^4^ PFs or RFs were seeded into each well while 5 × 10^4^ AN, M14, UM-SCC19 or UM-SCC47 were seeded into the upper insert simultaneously. Cells were then cultured at 37 °C in 5% CO_2_ for another 4 days.

### 4.3. Generation of Full-Thickness Models (FTMs)

Dermal matrices containing PFs or RFs were generated as described earlier [19]. Subsequently, 5 × 10^4^ HNSCC cells (UM-SCC19 or UM-SCC47) or 1.7 × 10^5^ primary keratinocytes were seeded onto the dermal matrix. FTMs were incubated for two days under submerged conditions at 37 °C with 7.3% CO_2_ in standard keratinocyte medium. Thereafter the culture medium was supplemented with 12 µM BSA, 5 µM β-dextrine, 10 µM L-carnitine, 10 mM L-serine, 50 nM selenious acid, 250 µM L-ascorbic acid phosphate, 1 µM DL-α-tocopherol, 7 µM arachidonic acid, 15 µM linoleic acid and 25 µM palmitic acid (all from Sigma-Aldrich) while the FBS was reduced to 1% and the hydrocortisone to 0.55 µM. After one day, 1 × 10^5^ melanoma cells (AN or M14) were either seeded onto the keratinocyte layer in submerged conditions or not (control group) and were cultured for another day for better cell localization. All FTMs were cultured at the air–liquid interface for 14 days using the culture medium described above, except that FBS was omitted and the linoleic acid concentration was increased to 30 µM. Culture medium was refreshed twice a week.

### 4.4. Masson Trichrome Staining

The Masson trichome staining was performed on 5 μm normal skin paraffin sections by using the Masson’s Trichrome Stain Kit (Polysciences Inc, #25088-100, Baden-Baden, Germany) according to the manufacturer’s protocol. Briefly, after rehydration of the paraffin sections, 100 μL Bouin’s solution was added on top of each sample for an hour at 60 °C. Slides were then washed in running tap water and then stained with Weigert’s iron hematoxylin, followed by a further incubation with Biebrich Scarlet-Acid Fuchsin Solution for 5 min. After 4 times of washing by distilled water, a 10 min incubation of phosphomolybdic acid was performed. Then, all samples were drained and stained with Aniline Blue for 2 min followed by a 1 min of Acetic acid incubation. Finally, all sections were dehydrated, mounted and visually inspected under a microscope.

### 4.5. Morphological and Immunohistochemical Analysis

From the FTMs, one part was fixed in 4% paraformaldehyde, dehydrated and paraffin embedded. Histological analysis was performed on 5 μm sections through hematoxylin and eosin (H&E) staining while immunohistochemical analysis was performed on 5 µm sections that were deparaffinized and rehydrated. For antigen retrieval, sections were boiled in citrate buffer (pH 6.0) in an autoclave at 110 °C for 5 min (Ki67, K17, K10, Vimentin, α-SMA, β-catenin, ZEB1 and SPARC), or incubated with 0.025% protease-X (Sigma-Aldrich) at 37 °C for 40 min (COL4) or autoclaved in Tris-EDTA buffer for 5 min (pH 9.0) (ZEB2 and COL11A1). After blocking in 2% Normal Human Serum (NHS), sections were incubated overnight with the corresponding primary antibodies at 4 °C and labelled using a streptavidin-biotin-peroxidase system (GE Healthcare), according to the manufacturer’s instructions. Staining was visualized with 3-amino-9-ethylcarbazole (AEC), counterstained with hematoxylin and mounted with Kaiser’s glycerin. For immunofluorescence (IF) staining, sections were labelled with desired primary antibodies as described above and counterstained with DAPI. Primary antibodies are specified in Appendix A.

### 4.6. Quantification of Relative Tumor Ingrowth

By using Image J, pictures of the H&E staining were first converted to grayscale. Tumor parts which broke through the basement membrane and entered into the dermal compartments were regarded as ingrowth area. For each FTM, the whole area of the dermis was calculated as “a”. Subsequently, boundaries were set manually using the line tool in Image J along the invaded segments of each FTM. The area of invaded tumor parts was calculated as “b”. The ratio of invaded tumor area in the whole dermis area in each FTM was then calculated as relative tumor ingrowth (b/a%). Results were expressed as the mean ± standard deviation (SD) of three counts from three different donors.

### 4.7. RNA Isolation and qPCR

For the 2D cell culture, a total RNA purification mini kit (Favorprep, #FATRK001-2, FAVORGEN, Vienna, Austria) was used for RNA isolation according to the manufacturer’s protocol. For the 3D-FTMs, one part of each FTM including dermis and epidermis was cut into small pieces and dissolved in lysis buffer containing 1.5% β-mercaptoethanol (Sigma-Aldrich, Zwijndrecht, the Netherlands). RNA isolation was performed using an RNeasy kit (Qiagen, Venlo, The Netherlands). cDNA conversion was performed using an iScript cDNA synthesis kit (BioRad, Veenendaal, The Netherlands) for all samples while an RT2 first strand kit was used specially to convert cDNA for the EMT kit (Qiagen), after which the expression of 84 genes was profiled with an EMT RT2 Profiler PCR Array (Qiagen) according to the protocol of the manufacturer. All primers were designed and evaluated with the amplification efficiency (determined by a dilution range of cDNA) and specificity (determined by gel electrophoresis). For all qPCR reactions, an IQ SYBR Green Supermix (BioRad) was used. qPCR was performed by using a CFX384 system (BioRad) according to the PCR program as described before [19]. The reference genes were selected based on stability using the Genorm program. The expression analysis was performed within the BioRad Software (CFX manager). Primers used for qPCR are specified in Appendix A.

### 4.8. Western Blotting

PFs and RFs were seeded in a 10 cm culture dish and cultured until the cell confluence reached 80% at 37 °C. M-PER (Thermo Scientific, Breda, The Netherlands) with a protease and phosphatase inhibitor cocktail (Thermo Scientific), which was used to extract total proteins. A Bradford assay was applied to determine the protein concentration. A total of 7 μg of protein samples were separated by Biorad miniprotein TGX gels at 80 V for 15 min, then at 100 V for 1 h. The proteins were further transferred to Polyvinylidene Difluoride (PVDF) membranes and blocked with 5% skim milk in Tris buffered saline with 0.1% Tween-20 for 1 h at room temperature, followed by incubating with the following different primary antibodies overnight at 4 °C: CLEC3B (Abcam, #ab51883, Amsterdam, The Netherlands) and β-actin (CST, #4987). The membranes were subsequently incubated with HRP-conjugated secondary antibody (1:2500) at room temperature for 1 h. Immunoblots were finally visualized by ECL (Pierce Scientific, Etten-Leur, The Netherlands) and recorded on X-ray film.

### 4.9. Statistical Analysis

GraphPad Prism software (version 8.0.1) was used to perform statistical analysis. Dual comparisons were made with an unpaired Student’s *t*-test. The results of three independent experiments were presented as mean ± standard deviation (SD), and *p* < 0.05 was considered statistically significant.

## 5. Conclusions

In this study, we explored the heterogeneity of PFs and RFs and for the first time showed the functional link between these two human fibroblast subtypes and the invasive behavior of melanoma and HNSCC. The PFs did not favor tumor invasion while the RFs formed a tumor-promoting environment by inducing EMT. Furthermore, the RFs tended to differentiate into CAFs when interacting with tumor cells. This study enriches our knowledge of the functional difference between papillary and reticular fibroblasts and adds evidence to our findings that PFs and RFs have distinct effects on tumor invasion in melanoma and HNSCC.

## Figures and Tables

**Figure 1 ijms-23-11651-f001:**
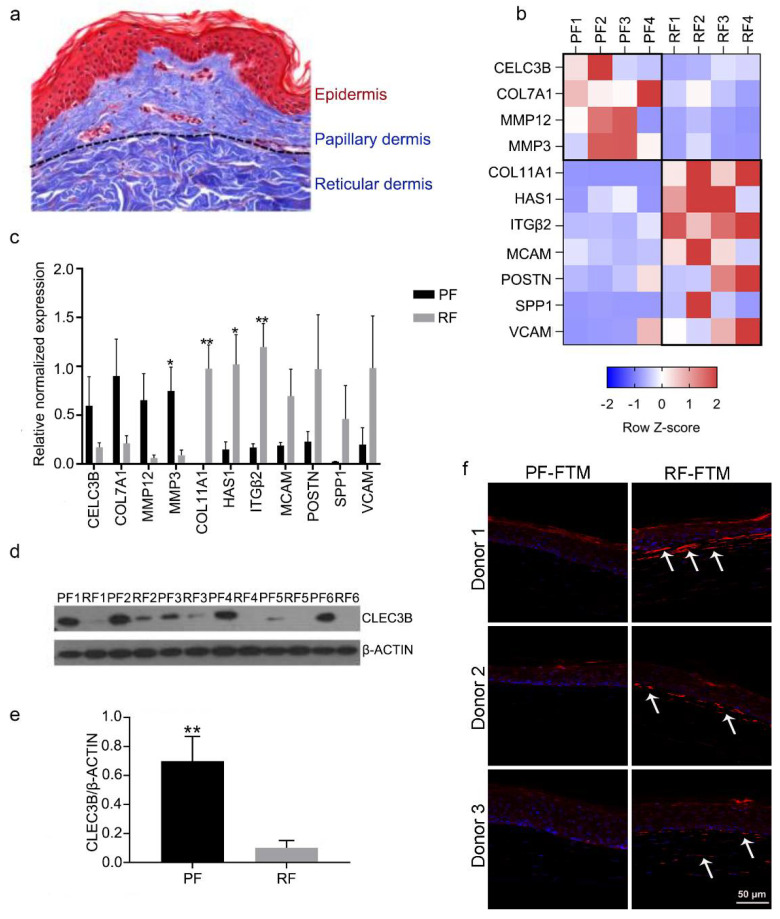
Distinct expression profiles of PFs and RFs. (**a**) Masson trichrome staining of normal human skin. (**b**) Heatmap of 11 differentially expressed genes in PFs and RFs on mRNA level, N = 4. (**c**) The mRNA expression of 11 differentially expressed genes in PFs and RFs, N = 4. (**d**) The protein expression of CLEC3B in PFs and RFs by Western blot, N = 6. (**e**) The Western blot quantification of CLEC3B and β-ACTIN. (**f**) The expression of COL11A1 in PF-FTMs and RF-FTMs by IF staining, N = 3. Positive staining is indicated by arrow as an example. Data is presented as mean ± standard deviation (SD); * *p* < 0.05, ** *p* < 0.01.

**Figure 2 ijms-23-11651-f002:**
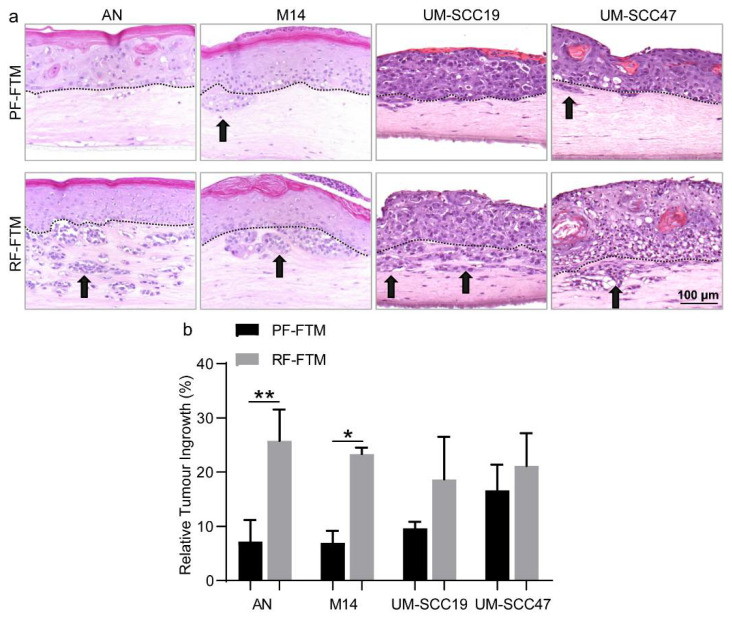
Effects of fibroblast subpopulations on melanoma and HNSCC invasion. (**a**) HE staining of PF-FTMs and RF-FTMs with different epidermis generated using AN, M14, UM-SCC19 or UM-SCC47 cell lines. Arrows indicate the invaded tumor area; dashed lines indicate the basement membrane separating epidermis and dermis. By using Image J, (**b**) quantification of the relative tumor ingrowth in AN, M14, UM-SCC19 and UM-SCC47-FTMs. Data was collected from three independent experiments and is presented as mean ± standard deviation (SD); * *p* < 0.05, ** *p* < 0.01.

**Figure 3 ijms-23-11651-f003:**
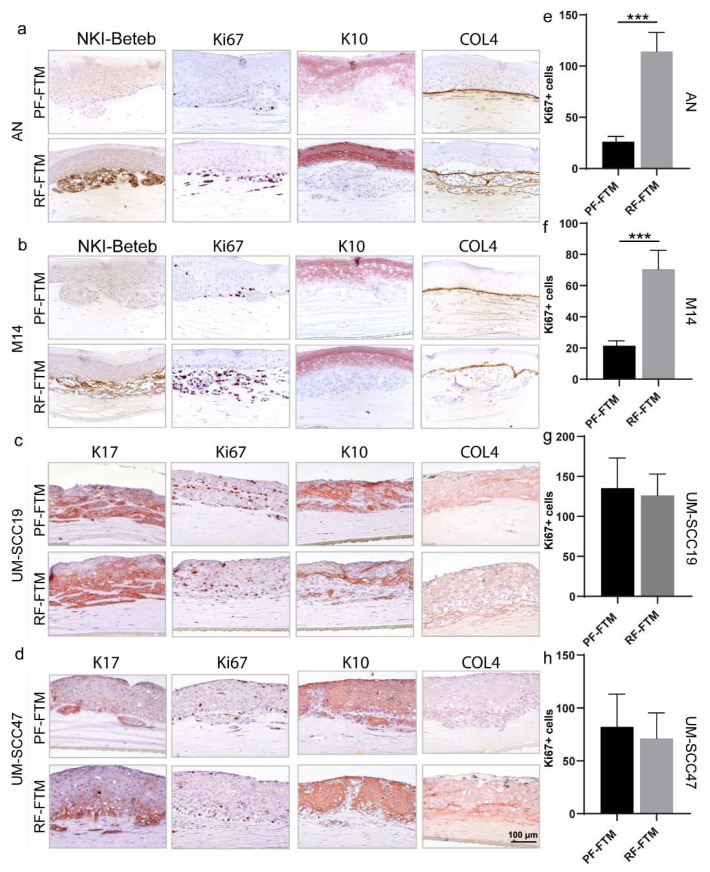
Effect of fibroblast subpopulations on melanoma and HNSCC morphogenesis. By IHC staining, the expression of NKI/Beteb, Ki67, K10 and COL4 were detected in (**a**) AN-FTMs and (**b**) M14-FTMs. The expression of K17, Ki67, K10 and COL4 was detected in (**c**) UM-SCC19-FTMs and (**d**) UM-SCC47-FTMs. The number of absolute Ki67-positive cells in (**e**) AN-FTMs, (**f**) M14-FTMs, (**g**) UM-SCC19-FTMs, and (**h**) UM-SCC47-FTMs was counted. Data was collected from three donors and from three independent experiments and is presented as mean ± standard deviation (SD); *** *p* < 0.001.

**Figure 4 ijms-23-11651-f004:**
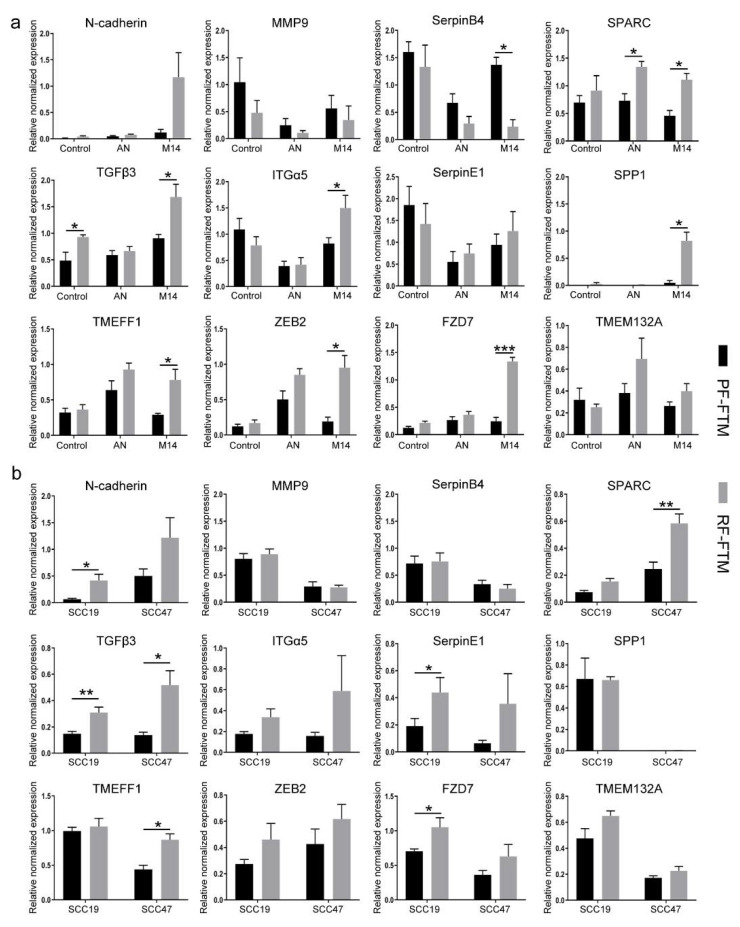
qPCR analysis of the EMT markers in melanoma or HNSCC FTMs. (**a**) Melanoma FTMs. (**b**) HNSCC FTMs. Data was collected from three independent experiments and is presented as mean ± standard deviation (SD); * *p* < 0.05, ** *p* < 0.01, *** *p* < 0.001.

**Figure 5 ijms-23-11651-f005:**
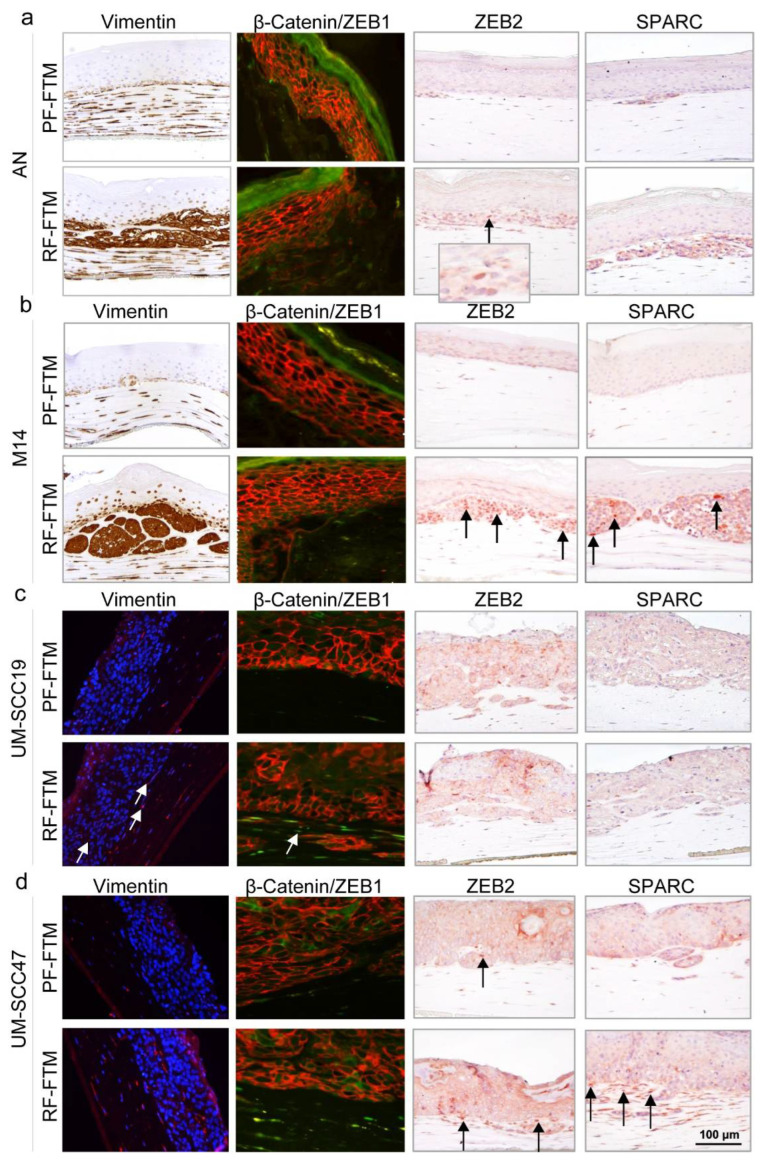
IHC and IF staining for EMT-related markers in melanoma or HNSCC FTMs. Vimentin, β-catenin/ZEB1, ZEB2 and SPARC were stained in PF-FTMs and RF-FTMs with (**a**) AN, (**b**) M14, (**c**) UM-SCC19 and (**d**) UM-SCC47 epidermis. Arrows indicate the expression of these biomarkers. Data was obtained from three independent experiments.

**Figure 6 ijms-23-11651-f006:**
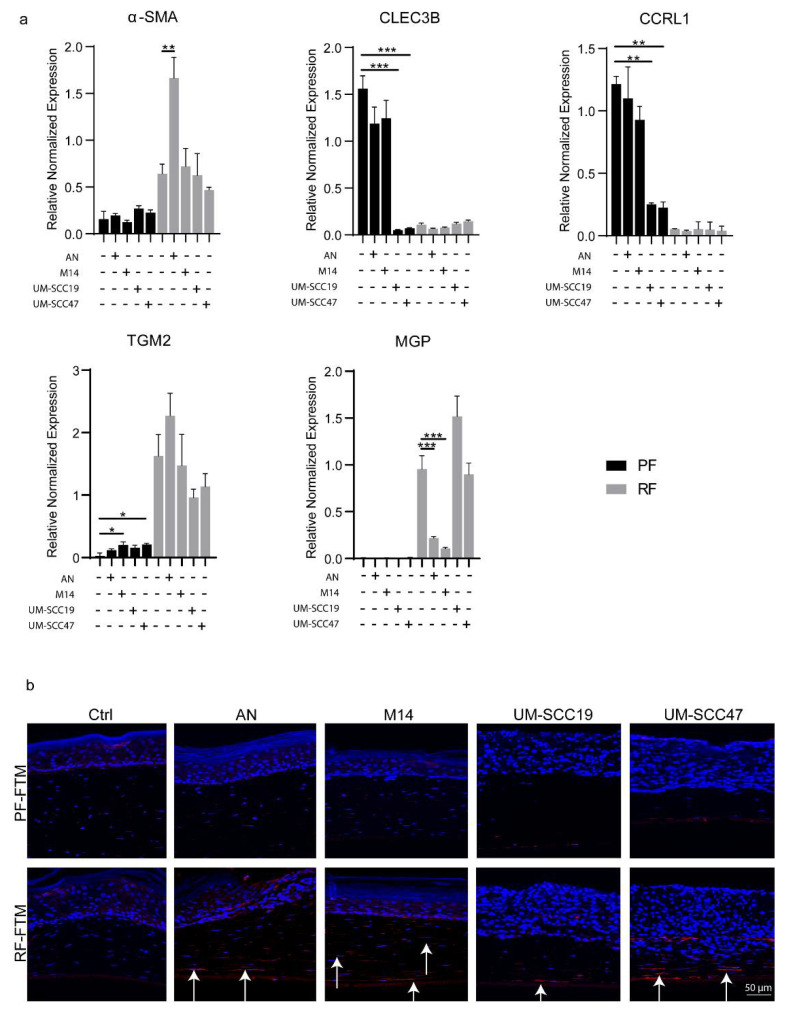
RF started to receive CAF phenotype after interaction with tumor cells. (**a**) The mRNA expression of α-SMA, CLEC3B, CCRL1, TGM2 and MGP in PFs and RFs co-cultured with AN, M14, UM-SCC19 or UM-SCC47. (**b**) α-SMA was stained in PF-FTMs and RF-FTMs with primary keratinocytes, AN, M14, UM-SCC19 or UM-SCC47 epidermis. Arrows indicate the expression of the biomarker. Data was obtained from three independent experiments and is presented as mean ± standard deviation (SD); * *p* < 0.05, ** *p* < 0.01, *** *p* < 0.001.

## Data Availability

The original contributions presented in the study are included in the article/Appendix A. Further inquiries can be directed to the corresponding author.

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
