# Peer review of "Human Papillary and Reticular Fibroblasts Show Distinct Functions on Tumor Behavior in 3D-Organotypic Cultures Mimicking Melanoma and HNSCC"

_ijms, 2022, doi:10.3390/ijms231911651_

Round 1
Reviewer 1 Report
The Authors conducted a study to explore the involvement of two different cell types, reticular (RF) and papillary fibroblasts (PF). Their work was based on previous studies, that showed a different molecular phenotype of these 2 cell types present in the skin. They show in 2D and 3D preclinical human tumor models, that HNSCC and melanoma models have more invasive properties and EMT-like phenotypes when cultured with RFs than with PFs.
Overall the study is well conducted and adds value to the scientific community.
However, there are a couple of things that should be addressed
1. Line 37. The pattern of metastatic spread of cancer dates back to 1889 when Steven Paget published his "seed and soil" hypothesis. Please correct the date in that sentence.
2.1 The method of measuring the invasion into the FTM needs to be described in more detail. At this point, the paragraph in Material and Methods is too short and comprehensive.
2.2 Figure 2 would benefit from a line indicating the "baseline" of non-invasion and an indicated area that is considered an invasion.
3. Line 360 has a typo. I believe it should be 4 x 104, not 104.
Author Response
Thank you for your detailed and helpful comments, we made following changes to the manuscript according to your suggestions:
1. The year was corrected into 1889 in the manuscript.
2.1. More details were added in the corresponding section in Material and Method.
2.2. Dashed lines and arrows were added in Figure 2 to better indicate the invaded tumor area.
3. The typo was corrected in Line 360.
Reviewer 2 Report
The manuscript by Wu et. al. describes the differential roles of papillary fibroblasts and reticular fibroblasts in regulating proliferation, invasion, and EMT features of melanoma and head and neck squamous cell carcinoma cell lines. Reticular fibroblasts were shown to promote higher level of tumor proliferation and invasion compared to papillary fibroblasts.
The manuscript is well written, clear, concise with well prepared figures and up to date references. The evidence supports the claims made in the manuscript. Only some minor comments need to be addressed.
1. In Figure 3, quantification of NKI-Beteb, K10, COL4, and K17 are missing. Please provide these quantification and differences between each fibroblast group.
Author Response
Thank you for your approval and suggestions. We made following changes to the manuscript according to your advice:
We added the quantification of NKI-Beteb, K17 and K10 and put the data in supplementary Figure 2 (Figure S2) due to limited space in Figure 3.
However, we did not perform the quantification of COL4 since our focus was not to observe the quantity of such protein between different groups but the appearance and structure disruption of the basement membrane.
I hope I explained it well and thanks again for your time.